# Clinical and Transcriptomic Characterization of Metastatic Hormone-Sensitive Prostate Cancer Patients with Low PTEN Expression

**DOI:** 10.3390/ijms26136244

**Published:** 2025-06-28

**Authors:** Marta Garcia de Herreros, Natalia Jiménez, Joan Padrosa, Caterina Aversa, Laura Ferrer-Mileo, Samuel García-Esteve, Leonardo Rodríguez-Carunchio, Isabel Trias, Laia Fernández-Mañas, Mercedes Marín-Aguilera, Mariana Altamirano, Manuel Mazariegos, Albert Font, Alejo Rodriguez-Vida, Miguel Ángel Climent, Sara Cros, Isabel Chirivella, Mariona Figols, Núria Sala-González, Vicenç Ruiz de Porras, Juan Carlos Pardo, Aleix Prat, Òscar Reig, Begoña Mellado

**Affiliations:** 1Translational Genomics and Targeted Therapeutics in Solid Tumors Lab, Fundació de Recerca Clínic Barcelona—Institut d’Investigacions Biomèdiques August Pi i Sunyer (FRCB-IDIBAPS), 08036 Barcelona, Spain; garciadehe@clinic.cat (M.G.d.H.); najimenez@recerca.clinic.cat (N.J.); padrosa@clinic.cat (J.P.); aversa@clinic.cat (C.A.); laferrer@clinic.cat (L.F.-M.); sgarciaes@recerca.clinic.cat (S.G.-E.); lafernandezm@clinic.cat (L.F.-M.); mmarin1@recerca.clinic.cat (M.M.-A.); mazariegos@recerca.clinic.cat (M.M.); alprat@clinic.cat (A.P.); 2Medical Oncology Department, Hospital Clínic, 08036 Barcelona, Spain; altamira@clinic.cat; 3Uro-Oncology Unit, Hospital Clínic, University of Barcelona, 08036 Barcelona, Spain; lerodrig@clinic.cat (L.R.-C.); itrias@clinic.cat (I.T.); 4Department of Medicine, University of Barcelona, 08036 Barcelona, Spain; 5Centro de Investigación Biomédica en Red de Enfermedades Raras (CIBERER), Instituto de Salud Carlos III, 28029 Madrid, Spain; 6Department of Pathology, Hospital Clínic, 08036 Barcelona, Spain; 7Medical Oncology Department, Institut Català d’Oncologia, Hospital Germans Trias i Pujol, 08916 Badalona, Spain; afont@iconcologia.net (A.F.); jcpardor@iconcologia.net (J.C.P.); 8Medical Oncology Department, Hospital del Mar, 08003 Barcelona, Spain; arodriguezvida@psmar.cat; 9Medical Oncology Service, Instituto Valenciano de Oncología (IVO), 46009 Valencia, Spain; macliment@fivo.org; 10Medical Oncology Department, Hospital General de Granollers, 08402 Barcelona, Spain; scros@fphag.org; 11Department of Medical Oncology, Hospital Clínico Universitario de Valencia, INCLIVA, University of Valencia, 46010 Valencia, Spain; chirivella_isa@gva.es; 12Medical Oncology Department, Fundació Althaia, Xarxa Assistencial Universitària de Manresa, 08242 Manresa, Spain; mfigols@althaia.cat; 13Department of Medical Oncology, Institut Català d’Oncologia, 17007 Girona, Spain; nsgonzalez@iconcologia.net; 14Badalona Applied Research Group in Oncology (B-ARGO), Institut Català d’Oncologia, 08916 Germans Trias i Pujol Badalona, Spain; vruiz@igtp.cat

**Keywords:** hormone-sensitive prostate cancer, PTEN, biomarkers, androgen deprivation therapy, docetaxel, androgen receptor signaling inhibitors, CHAARTED trial

## Abstract

Alterations in the *PTEN* tumor suppressor gene are common in prostate cancer. They have been associated with a more aggressive disease and poor outcomes and potential benefit of targeted therapies. The purpose of this work is to study the clinical and transcriptional landscapes associated to low *PTEN* mRNA expression in metastatic hormone-sensitive prostate cancer (mHSPC) patients. A multicenter biomarker ambispective study was performed in mHSPC patients. *PTEN* mRNA expression was assessed by nCounter in formalin-fixed paraffin-embedded tumor samples. PTEN_low_ status was defined by a previously validated cut-off and was correlated with castration-resistant prostate cancer (CRPC)-free survival (CRPC-FS) (primary endpoint) and overall survival (OS). RNA-Seq was performed to molecularly characterize PTEN_low_ vs. PTEN_wt_ tumors. A total of 380 patients were included, 350 eligible. PTEN_low_ was observed in 28.2% of patients and was independently associated with shorter CRPC-FS (HR 1.6, 95% CI 1.2–2.1, *p* = 0.002) and OS (HR 1.5, 95% CI 1.1–2, *p* = 0.014). PTEN_low_ tumors showed overexpression of neuroendocrine, cell cycle, and DNA repair gene signatures, reduced expression of the androgen receptor pathway, and a distinct immune microenvironment. Using microarray data from the CHAARTED trial, we developed a PTEN-low related signature, independently associated with CRPC-FS (HR 1.5, 95% CI 1–2.3, *p* = 0.036) and OS (HR 1.9, C1 1.2–2.9, *p* = 0.005), and identified targets for potential therapies in PTEN-altered tumors. We conclude that PTEN_low_ correlates with an aggressive clinical outcome in mHSPC patients and is associated with a unique transcriptional profile. These findings further support the investigation of novel therapeutic strategies for patients with *PTEN* alterations.

## 1. Introduction

The survival of patients with metastatic prostate cancer (PC) has significantly improved in recent years due to the introduction of combination therapies, including androgen deprivation therapy (ADT) with docetaxel (D), androgen receptor signaling inhibitors (ARSI), or triplet therapy (ADT + ARSI + D) as first-line treatment for metastatic hormone-sensitive prostate cancer (mHSPC) [1,2,3,4,5,6,7,8].

Despite the lack of biomarkers to effectively guide clinical decisions in mHSPC patients, recent research suggests that alterations in the tumor suppressor genes phosphatase and tensin homolog (*PTEN*), *RB1*, and *TP53* are associated to an aggressive clinical course [9,10]. This highlights the need for a deeper understanding of the molecular landscape and the development of distinct treatment strategies for this subgroup of patients. Notably, promising results have recently been reported for the combination of abiraterone in PTEN-deficient de novo metastatic mHSPC patients [11]. *PTEN* is a key tumor suppressor in PC, encoding a dual-specificity phosphatase that negatively regulates the PI3K/AKT signaling pathway [12]. Functioning as a tumor suppressor, *PTEN* exerts its influence by dephosphorylating phosphatidylinositol (3,4,5)-trisphosphate (PIP3), thereby antagonizing the pro-survival and pro-growth signals mediated by the PI3K pathway that regulate a wide range of cellular processes such as growth, proliferation, survival, motility, and metabolism [12,13]. PTEN loss, observed in 20–40% of PC, encompasses a high variety of alterations, such as genomic deletions of either one or both copies of the *PTEN* gene, loss of function mutations, epigenetic modulation, and post-translational modifications that lead to reduced PTEN levels [14]. The final consequence is an aberrant activation of the PI3K/AKT pathway enhancing PC proliferation and preventing apoptosis [13]. Several studies indicate that the loss of *PTEN* along with the aberrant activation of *PI3K/AKT* pathway could be implicated in ARSI and taxane resistance, particularly within the context of castration-resistant prostate cancer (CRPC) [15,16].

Alterations in *PTEN* become more frequent as the disease progresses, with rates around 15–20% in localized tumors, 30–40% in mHSPC, and up to 50% in advanced CRPC [9,14,17]. In all three settings PTEN loss has been associated with worse outcomes and decreased survival rate [9,10,14,17,18,19,20]. In our previous work, we showed that low *PTEN* mRNA expression in tumor samples detected by nCounter correlated with *PTEN* mutations and protein expression assessed by immunohistochemistry (IHC), as well as with poor clinical outcomes of mHSPC patients [10]. The adverse prognostic value of *PTEN* mRNA expression was also reported in a large cohort of patients from the STAMPEDE trial, where PTEN loss signatures were prognostic for shorter overall survival (OS) [21] but not predictive of abiraterone benefit. In the present work, we validate the prognostic role of *PTEN* expression in a larger series of mHSPC patients. Additionally, we performed a transcriptional characterization of PTEN_low_ tumors and developed a PTEN-low related gene signature that also predicts adverse clinical outcomes in mHSPC. We discuss the more relevant transcriptional findings and propose potential therapeutic interventions.

## 2. Results

### 2.1. Patients and Samples

A total of 380 patients were included in this study: 355 were eligible and 25 (7%) were excluded because of insufficient tumor sample (*n* = 11) or lack of RNA availability (*n* = 14). Most formalin-fixed paraffin-embedded (FFPE) samples were obtained from the primary tumor (95.5%) and the remainder from metastatic sites. Of the included patients, 125 were treated with ADT + D, 137 with ADT + ARSI, and 93 with ADT alone. The group of patients treated with ADT + D presented higher levels of prostate-specific antigen (PSA) at diagnosis (*p* < 0.001) and higher frequency of de novo stage IV (92.8%, *p* < 0.001) and high-volume disease (77.6%, *p* < 0.001) (Appendix A). The median follow-up was 41.4 months (2.4–223.5), 241 patients (67.9%) developed CRPC, and 219 died (61.7%). Median CRPC-free survival CRPC-FS and OS were 23.9 (95% CI 20.6–28.9) and 49.1 months (95% CI 43.9–58.1), respectively. Appendix A separately shows the clinical characteristics, median follow-up time, the rate of CRPC, and death per treatment group.

### 2.2. PTEN Expression and Clinical Evolution

Overall, 100 patients (28.2%) were considered PTEN_low_ using our previously validated expression cut-off (33.3% in ADT, 28% in ADT + D, and 24.8% in ADT + ARSI cohorts). PTEN_low_ patients did not exhibit significant differences in clinical or biological characteristics (Gleason, PSA, or lactate dehydrogenase [LDH] levels) with respect to those with PTEN_wt_. Table 1 and Appendix A summarize clinical characteristics of all patients and each treatment cohort by *PTEN* status, respectively.

In our series, PTEN_low_ was associated with shorter CRPC-FS (16.6 vs. 29.3 months; Hazard ratio [HR] 1.7, 95% confidence interval [CI] 1.3–2.2, *p* < 0.001) and OS (40.1 vs. 57.9 months; HR 1.5, 95% CI 1.2–2.1, *p* = 0.003) (Figure 1A–D). In the multivariate analysis, PTEN_low_ was independently associated with lower CRPC-FS (HR 1.6, 95% CI 1.2–2.1, *p* = 0.002) and OS (HR 1.5, 95% CI 1.1–2, *p* = 0.014) (Figure 1E,F).

Regarding each treatment series separately, PTEN_low_ was correlated with shorter CRPC-FS and OS in the ADT cohort (14.6 vs. 18.6 months; HR 1.7, 95% CI 1.1–2.8, *p* = 0.019; 41.4 vs. 49.1 months; HR 1.6, 95% CI 1–2.6, *p* = 0.042, respectively) and in the ADT + D cohort (12.1 vs. 21.9 months; HR 1.8, 95% CI 1.2–2.8, *p* = 0.006; 38.8 vs. 60.1 months; HR 1.7, 95% CI 1–2.7, *p* = 0.032, respectively). In the ADT and ADT + D cohorts, PTEN_low_ was independently associated with shorter CRPC-FS (HR 1.6, 95% CI 1–2.6, *p* = 0.047; HR 1.7, 95% CI 1.1–2.8, *p* = 0.029, respectively) (Appendix A). The ADT + ARSI cohort, with fewer events, exhibited a trend towards shorter CRPC-FS (Appendix A). The interaction test showed no significant interaction between PTEN_low_ and treatment for CPRC-FS (*p* = 0.966 and *p* = 0.631 for ADT + D and ADT + ARSI vs. ADT, respectively) and OS (*p* = 0.99 and *p* = 0.61 for ADT + D and ADT + ARSI vs. ADT, respectively).

To validate our results, we analyzed microarray data from 160 patients included in CHAARTED clinical trial as an independent cohort [19] where 53 patients (32.7%) were classified as PTEN_low_. As observed in our series, in the univariate analysis, PTEN_low_ patients had worse CRPC-FS (10.7 vs. 15.8 months; HR 2.2, 95% CI 1.1–4.3, *p* = 0.033) and OS (31.6 vs. 48.8 months; HR 2.2, 95% CI 1–4.5, *p* = 0.039) than PTEN_wt_ patients (Figure 2A–D). However, PTEN status was not independently associated to outcome in this series. PTEN_low_ patients treated with ADT + D presented longer CRPC-FS compared to those treated with ADT alone (15.2 vs. 6.2 months; HR 1.9, 95% CI 1.2–3.6, *p* = 0.034) (Figure 2E,F). However, PTEN_wt_ patients treated with ADT + D presented a significant longer CRPC-FS (23.9 vs. 9.9 months; HR 2.2, 95% CI 1.4–3.4, *p* < 0.001) and OS (53.9 vs. 44 months; HR 1.7, 95% CI 1–2.9, *p* = 0.038) compared to those treated with ADT alone (Figure 2G,H).

### 2.3. Transcriptional Characterization of PTEN_low_ Tumors

In order to transcriptionally characterize PTEN_low_ tumors, we performed an RNA-Seq analysis of tumor samples from 60 patients treated with ADT + D [10]. Clinical characteristics of these patients are summarized at Appendix A.

PTEN_low_ tumors were characterized by overexpression of genes associated with cell cycle, chromosomal instability, and tumor cell invasion transcripts (Figure 3A and Appendix A). Of note, from these DEGs, we identified eight genes that codify membrane proteins, nine known to be cancer-related, and four with available targeted inhibitors (Figure 3B).

In GSEA we identified significantly upregulated pathways in PTEN_low_ tumors, including the estrogen receptor pathway, glycolysis metabolism pathway, E2 F-cell cycle pathway, DNA repair, cell cycle pathways, and IL2-mediated inflammatory response. The only significantly downregulated pathway in PTEN_low_ tumors was the androgen receptor (AR) pathway (*p*-adj = 0.032) (Figure 3C).

When assessed the overenriched pathways in PTEN_low_ samples from the CHAARTED trial microarray dataset, and when compared to our findings, the GSEA shared differentially expressed pathways in the PTEN_low_ population, including glycolysis, TP53, IL2, and DNA repair pathways, reinforcing the results observed in our patient cohort (Figure 3D).

### 2.4. Development of a PTEN-Low Related Signature

To capture the true “PTEN-low” phenotype, we applied a cross-validation approach using microarray data from the CHAARTED clinical trial [19]. A total of 143 microarrays were deemed of sufficient quality to perform the analysis. An elastic-net model identified a 39-gene signature defining PTEN_low_ status (Figure 4A). This gene signature yielded a mean area under the curve (AUC) of 0.75 and showed a significant correlation with PI3K/AKT/mTOR, IL2, and interferon alpha pathways (FDR < 0.05) (Appendix A). In this training cohort, PTEN-low signature expression was independently associated with shorter CRPC-FS (HR 1.5, 95% CI 1–2.3, *p* = 0.036) and OS (HR 1.9, C1 1.2–2.9, *p* = 0.005) (Figure 4B,C). The 39-gene set was then scored per sample in the RNA-Seq dataset, as a validation cohort (*n* = 60), with an AUC of 0.59 for classifying PTEN_low_. No significant association was observed with CRPC-FS or OS. However, the signature correlated with PI3K/AKT/mTOR, interpheron alpha, and IL2-STAT5 patwhays, recapitulating the findings of the training cohort (Appendix A).

### 2.5. Immune Microenvironment-Related Gene Expression in PTEN_low_ Tumors

We next focused on PTEN_low_ immune microenvironment gene expression to assess if PTEN_low_ tumors are characterized by an immunosuppressive microenvironment, as previously suggested [22,23]. We found that PTEN_low_ tumors presented an upregulation of inflammatory-response-related genes, both the innate and adaptive immune system (Figure 5A). Regarding the innate immune response, PTEN_low_ tumors exhibited higher levels of neutrophils and M2 macrophages (*p* = 0.012 and *p* = 0.002, respectively) but lower myeloid dendritic cells (*p* = 0.007). Within the adaptive immune system, PTEN_low_ tumors showed a trend toward higher expression of effector CD8+ cells and B lymphocytes, with overexpression of the immunoglobulin gene signature (IGG signature) [24], Zhao’s prostate immune-related signature [25], and exhausted T cell signature. Conversely, PTEN_low_ tumors displayed lower expression of activated CD4+ T cells and the gene expression profile (GEP) signature, associated with T cell inflammation and immune activation (Figure 5A,B).

### 2.6. Neuroendocrine Gene Expression in PTEN_low_ Tumors

In the GSEA analysis from RNA-Seq data, we observed that PTEN_low_ tumors exhibited upregulation of the neuroendocrine (NE) signature defined by Beltran [26] (*p*-adj = 0.092). At the individual gene level within the NE signature, we observed a significant negative correlation between Enhancer of zeste homolog 2 (*EZH2*), *PTEN,* and other NE-related genes (Figure 6A). Consistently, in the nCounter data, PTEN_low_ tumors demonstrated significantly higher *EZH2* mRNA levels (Figure 6B,C). Based on these findings, and given the evidence suggesting that *EZH2* functions as a transcriptional repressor of *PTEN* [27], we further investigated the prognostic role of the co-expression of these two genes (nCounter expression). Patients with PTEN_low_-EZH2_high_ tumors (9.6%) experienced significantly shorter CRPC-FS (15.4 vs. 25.5 months; HR 1.8, 95% CI 1.2–2.6, *p* = 0.001) and OS (36 vs. 53.3 months; HR 1.9, 95% CI 1.3–2.8, *p* < 0.001) compared to patients with wild-type expression of both genes (Figure 6D,E and Appendix A). The multivariate analysis confirmed the correlation of PTEN_low_-EZH2_high_ with shorter CRPC-FS and OS (Figure 6F). Among the PTEN_low_ tumors, higher *EZH2* expression was significantly associated with shorter OS (36 vs. 48.9 months, HR 1.8, 95% CI 1.2–2.9, *p* = 0.038) (Appendix A).

## 3. Discussion

In this study, we show that low-*PTEN* mRNA expression (PTEN_low_) independently correlates with shorter time to CRPC and reduced OS in patients with mHSPC, irrespective of the up-front treatment received. While patients with PTEN_low_ tumors did not show significant differences in clinical characteristics compared to those with PTEN_wt_, they exhibited a distinct transcriptional landscape. PTEN_low_ tumors were characterized by underexpression of the AR pathway and overexpression of the PI3K/AKT/mTOR, glycolysis, DNA repair, and immune pathways. Moreover, we developed a 39-gene PTEN-low related signature that captured these molecular alterations and correlated to adverse outcomes in the CHAARTED trial patients.

The adverse prognostic significance of PTEN loss has been extensively studied in CRPC, particularly through analyses of *PTEN* mutations and reduced IHC expression [9,14,18]. However, far fewer studies have investigated the detrimental role of PTEN alterations in mHSPC and only one study has explored a PTEN loss signature in this setting [21], reporting results consistent with those presented here.

Some *PTEN* transcriptional signatures have been previously characterized by other authors. Imada et al. analyzed public PC datasets and derived a gene expression signature linked to PTEN inactivation [23], while Saal et al. developed a transcriptomic signature reflecting PTEN deficiency in breast cancer tumors [28]. In both signatures, PTEN loss was defined by reduced IHC expression, and each demonstrated hyperactivation of the PI3K/AKT/mTOR and cell cycle pathways, which correlated with adverse clinical outcomes. Liu and colleagues also developed a *PTEN* transcriptional signature consisting of 45 differentially expressed genes identified in PTEN-deleted PC tumors from public databases [29]. Both Liu and Saal signatures were validated in an exploratory analysis of the STAMPEDE trial, where they were associated with worse survival [21].

In the present study, we developed a 39-gene PTEN-low related transcriptomic signature using microarray data from CHAARTED mHSPC tumors by comparing mRNA expression profiles between PTEN_low_ and PTEN_wt_ samples. This classification was based on a previously validated cut-off that correlates with PTEN IHC expression and genomic alterations [10]. However, the use of a single *PTEN* mRNA expression cut-off may pose a limitation regarding transferability across different platforms and patient cohorts. By developing a multi-gene signature, we aimed to more accurately capture the true “PTEN low” phenotype, potentially offering greater biological and clinical relevance than relying on *PTEN* mRNA threshold alone. Our PTEN-low related signature effectively recapitulated key downstream biological features of PTEN loss, particularly the activation of the PI3K/AKT/mTOR pathway and immune-related programs such as IFN-α and IL-2/STAT5. Despite being developed without survival data, the signature predicted both CRPC-FS and OS in the training cohort, supporting its biological and potential clinical relevance. However, prognostic value was not replicated in the smaller RNA-Seq validation set, likely due to limited event numbers and low power. Further prospective validation is required to confirm the prognostic and biological value of the PTEN-low related signature in larger and independent patient series. Notably, several genes in our PTEN-low signature have been previously linked to cancer and overlap with those identified in PTEN-related signatures by Liu and Saal [28,29]. These genes are involved in critical processes like cell adhesion, motility, proliferation, and metastasis, which are known to drive poor outcomes across multiple cancer types [30]. Importantly, some of the enriched genes in PTEN_low_ tumors, such as *AKT3*, *CDC7*, *SLC16A3*, and *BIRC5*, have inhibitors currently under development, highlighting their potential as promising targets for therapeutic intervention in PTEN_low_ tumors [31,32,33,34]. In the GSEA, we found a correlation between low *PTEN* expression and critical pathways in PC biology, including upregulation of the estrogen receptor pathway, glycolysis, DNA repair, cell cycle, and NE signatures, along with a lower expression of the AR pathway. The elevated DNA repair signature may be attributed to the link between PTEN loss and chromosomal instability, which activates DNA damage repair pathways, such as ATM/ATR [35]. Our findings suggest that this pathway could represent a potential therapeutic target with PARP or ATR inhibitors, as previously explored in PTEN-altered breast cancer or glioblastoma [36,37].

The lower expression of the AR receptor pathway found in PTEN_low_ tumors is consistent with previous evidence [10], where AR transcriptional activity is reduced in PTEN-null tumors and AR inhibition activates AKT signaling through reciprocal cross-talk [16]. This may explain why PTEN_low_ tumors have poorer response to hormonal treatments [18]. Preclinical studies have shown this reciprocal feedback whereby AR inhibition upregulates PI3K/AKT/mTOR pathway signaling and PI3K/AKT/mTOR pathway inhibition activates AR signaling [16]. Simultaneous inhibition of both pathways has shown preclinical antitumor activity, particularly in PTEN-deficient models [16]. The phase III IPATential150 trial demonstrated that AKT inhibitor, ipatasertib, improved radiological progression-free survival when added to abiraterone in mCRPC patients harboring *PTEN* mutations [38]. In mHSPC, the ongoing phase III CAPItello-281 study (NCT04493853) is evaluating the addition of capivasertib to abiraterone in mHSPC patients with PTEN-deficient tumors. Preliminary data favor the capivasertib regimen, although the results are still immature [39].

Consistent with prior reports, our analysis revealed increased expression of NE-related genes and epigenomic regulators, previously linked to PTEN deficiency [40]. Among them, one of the pivotal players is *EZH2*, a histone *methyltransferase* enzyme of the Polycomb repressive complex [41]. EZH2 is a master regulator of NE lineage plasticity in PC and has been associated with the methylation and silencing of *PTEN* in various tumor types [27]. We found that low *PTEN* expression and high *EZH2* expression were associated with a remarkable poor prognosis, suggesting that this subgroup of tumors (PTEN_low_—EZH2_high_), may need a better treatment strategy. Notably, EZH2 has shown promising activity in early trials with CRPC patients [42,43] and it is currently in investigation in mHSPC.

PTEN loss has been shown to affect the immune system in a PI3K/AKT-independent manner, thereby modulating the tumor immune microenvironment [22,44]. Previous reports across different cancer types, have shown that disrupted *PTEN* expression leads to a decreased immune response and increased immunosuppressive cytokines [45,46]. Vidotto et al. analyzed public cancer datasets and observed that PTEN-deficient tumors frequently exhibited dysregulation of immune-related pathways [44]. In most cancer types, PTEN loss was associated with a higher abundance of CD4+ lymphocytes, M1 macrophages, and FoxP3+ T regulatory cells, along with increased expression of immune checkpoints such as LAG3 and IDO1, indicating an immunosuppressive tumor microenvironment [44]. In contrast, a recent study found that prostate tumors with PTEN loss exhibited strong activation of both innate and adaptive immune systems, with increased interferon-gamma response genes and CD8+ lymphocytes, suggesting they may respond better to immunotherapies [23]. Likewise, in our study, we observed a significant dysregulation of most of these immune cell’s signatures in PTEN_low_ tumors. On one side, there was an overexpression of neutrophils, macrophages and fibroblasts, reinforcing the presence of an immunosuppressive microenvironment that could promote tumor progression and invasion. Interestingly, we also detected elevated CD8 +, B cell, and exhausted T cell signatures, suggesting that, although PTEN_low_ tumors may harbor a higher frequency of CD8+ T cells, these may be functionally exhausted due to the suppressive immune microenvironment. Given that PTEN_low_ tumors display reduced androgen production, which strongly suppresses inflammatory immune cells [47], we hypothesized that this reduction in androgen levels might trigger a more effective immune response. Additionally, the increased genomic instability associated with PTEN loss may enhance tumor antigenicity, thereby promoting immune activation via cGAS–STING signaling [35]. These findings are of particular interest as they open the door to exploring immunotherapy-based approaches in PTEN-altered prostate tumors, a setting traditionally considered less responsive to such treatments.

Our study does face limitations, including its retrospective and non-randomized design, which may be subject to selection bias and missing data, as well as the treatment heterogeneity across the first-line therapy groups, which may introduce residual confounding despite multivariable modeling and the lack of a cohort of patients treated with triple therapy (ADT+D+ARSI). Additionally, although the PTEN-low related signature demonstrated clear biological validity, its prognostic value remains unconfirmed. Larger RNA-Seq cohorts will be necessary to establish its clinical relevance. Finally, further research is warranted to better characterize the immune tumor microenvironment and its relationship with PTEN loss.

In conclusion, this study demonstrates in a large cohort of mHSPC patients that low *PTEN* expression is an adverse prognostic factor. Through comprehensive characterization of PTEN_low_ tumors and their immune microenvironment, we identified a prognostic PTEN-low related signature and potential targetable genes. Our findings offer new insights into the transcriptional landscape associated with low *PTEN* expression, which may aid in the identification of targeted therapies and support the development of personalized treatment strategies for patients with PTEN-altered tumors.

## 4. Materials and Methods

### 4.1. Design, Patients, and Samples

We present an exploratory multicenter ambispective ongoing biomarker study in patients with mHSPC in different hospitals from Spain. Key inclusion criteria were prostate adenocarcinoma diagnosis with available FFPE biopsy of the primary tumor or a metastatic site in the hormone-sensitive setting and enough material for molecular analysis as assessed by the pathologist. Treatment for mHSPC was ADT alone (i.e., luteinizing hormone-releasing hormone (LHRH) analogs), ADT in combination with D (75 mg/m^2^ every 21 days for six cycles), or ADT in combination with ARSI (enzalutamide 160 mg/day, apalutamide 240 mg/day, or abiraterone 1000 mg/day in combination with prednisone 5 mg/day). Patients with primary NE tumors were excluded. Clinical variables were collected from patient’s electronic records. The primary endpoint of the study was to correlate *PTEN* mRNA expression with CRPC-FS. Secondary endpoints were to correlate *PTEN* mRNA expression with OS, to molecularly characterize PTEN_low_ tumors and the different pathways associated with PTEN_low_, using a multiplatform approach (RNA-Seq and microarrays), and to develop a PTEN-low related transcriptomic signature to capture the PTEN-low biology. Considering our previous results [10] we estimated a sample size of 300 patients to detect differences in the primary endpoint with power = 0.8 and *α* = 0.05, two-tailed.

### 4.2. Gene Expression Panel Design

We have configured a gene expression nCounter panel (Nanostring Technologies, Seattle, WA, USA) of 184 genes representing signatures described to be related with CRPC development and androgen suppression or taxanes resistance [48]. Here, we present the data focused on *PTEN* and *PTEN*-related pathway genes.

### 4.3. RNA Extraction

Sections of FFPE tumor tissues (prepared as previously described [48]) were stained with hematoxylin and eosin to determine the tumor area. At least two 10 μm macrodissected FFPE slides were used to extract total RNA by using AllPrep DNA/RNA FFPE Kit (QIAGEN, Hilden, Germany) according to the manufacturer’s instructions. RNA was quantified by a Nanodrop Spectrophotometer ND-1000 (Thermo Scientific, Wilmington, MA, USA).

### 4.4. nCounter Gene Expression Analysis

A minimum of ~100 ng of total RNA was used to measure gene expression using the nCounter platform following the manufacturer’s protocol (Nanostring Technologies, Seattle, WA, USA). RNA was hybridized into 192 probe sets for 18 h at 67 °C and processed as previously described [48]. Here we present the data focused on *PTEN* and *EZH2.* Raw expression counts (Appendix A) were collected, normalized, and log2 transformed using the nSolver 4.0 software (RRID:SCR_003420) [48].

### 4.5. RNA Sequencing (RNA-Seq)

Sample quality control, sequencing library preparation, qualification, and pooling, as well as NGS data processing, were performed by the High Content Genomics and Bioinformatics Unit at the Institut Germans Trias i Pujol (IGTP). Pooled library gel purification and quality control and Illumina sequencing procedures were performed by the Genomics Unit at the Center for Genomic Regulation (CRG) [10].

### 4.6. Bioinformatics and Statistical Analysis

A previously established and validated cut-off by our group [10] was applied to transformed (z-score) *PTEN* nCounter gene expression data of each cohort (described in the Results section) to define PTEN_low_ tumors. PTEN_wt_ was considered for the remaining cases. This cut-off was also applied to transformed (z-score) microarray data from the CHAARTED trial patients [19]. For *EZH2*, tertiles were applied to transformed (z-score) nCounter gene expression data of all patients and EZH2_high_ was considered for patients with the higher tertile expression.

RNA-Seq data was normalized and a differential gene expression analysis was performed using DESeq2 (RRID:SCR_015687). Genes with false discovery rate (FDR) adjusted *p*-value (*p*-adj) < 0.1 and absolute log2 fold change (LFC) ≥ 1 were considered as statistically significant differentially expressed genes (DEGs). GSEA was performed with the fgsea package (RRID:SCR_020938) using as input the full list of genes ranked by transformed *p*-value (signed *p*-value = −log(*p*-value) × sign(LFC)). MSigDB Hallmarks [49], NE signatures [26], and different immune datasets were used to obtain pathways of interest [24,25,50]. Single-sample GSEA (ssGSEA) from the GSVA package (RRID:SCR_021058) was used to compare immune expression profiles between PTEN_low_ and PTEN_wt_.

We reanalyzed microarray data from 160 HSPC biopsies from the CHAARTED trial [19]. Raw Affymetrix HuEx-1_0-ST CEL files were processed using the oligo v1.64 (RRID:SCR_015729). Quality control was performed using arrayQualityMetrics v3.54 (RRID:SCR_001335). Expression data were background-corrected and quantile-normalized with Robust Multiarray Average (RMA), retaining ~22,000 transcript clusters. Probe sets without HGNC symbols were discarded, and batch effects were captured by scanning the chip date and carried downstream. For genes with multiple probe sets, the mean expression was calculated.

The PTEN-low related signature was developed using 10 × 10 repeated cross-validation on the CHAARTED microarray dataset [19], applying an elastic-net logistic regression model (*α* = 0.2) implemented with caret v6.0–94 (RRID:SCR_021138) and glmnet v4.1 packages (RRID:SCR_015505). Predictors were centered and scaled (mean = 0, SD = 1), and model performance was assessed by AUC. The final *λ* was selected using the 1-standard error rule to favor model simplicity while retaining predictive power. Model parameters were locked for external application. To overcome the intrinsic scaling difficulty of cross-platform validation (from microarray to RNA-Seq) when assessing the prognostic value of the signature, each sample was scored using GSVA v1.50 (RRID:SCR_021058) and classified based on the median score of the signature. In parallel, scores were also calculated for Hallmark gene sets (MSigDB v7.5), and their correlation with the PTEN-low signature score was assessed using Spearman correlation with FDR correction. For validation, the same methodology was applied to our RNA-Seq samples. ROC analysis was also performed to evaluate the PTEN-GSVA score’s ability to classify RNA-Seq samples as PTEN_low._

Clinical variables such as de novo stage IV, Gleason at diagnosis, the presence of visceral metastasis, bone metastasis, volume (as defined in CHAARTED trial [1]), and risk (as defined in LATITUDE trial [3]) at ADT start time were evaluated as dichotomic. PSA at diagnosis and LDH levels at ADT start time were evaluated as continuous variables.

CRPC-FS and OS were calculated from the date of start of ADT to the time of developing CRPC and to the time of death or last follow-up visit, respectively. Survival analyses were performed by the Kaplan–Meier method and compared by log-rank test. CRPC-FS definition, treatment–response criteria, and progressive-disease definitions followed Prostate Cancer Working Group 2 criteria [51]. Univariate and multivariate analysis were performed by Cox regression analysis.

Fisher’s exact test and the Wilcoxon Mann–Whitney test were used to compare the proportions of qualitative and continuous clinical variables between two groups, respectively. Chi-square test and Kruskal Wallis test were used to compare categorical and continuous variables between three groups, respectively.

A test of interaction was performed by entering proportional hazard models’ selected multiplicative interaction terms between two variables: treatment (ADT, ADT + D, or ADT + ARSI) and *PTEN* expression (PTEN_low_ or PTEN_wt_). The correlation between *PTEN* and the other relevant genes was assessed using RNA-Seq through Pearson correlation test. Analyses were performed with R software (v.4.3.2).

## Figures and Tables

**Figure 1 ijms-26-06244-f001:**
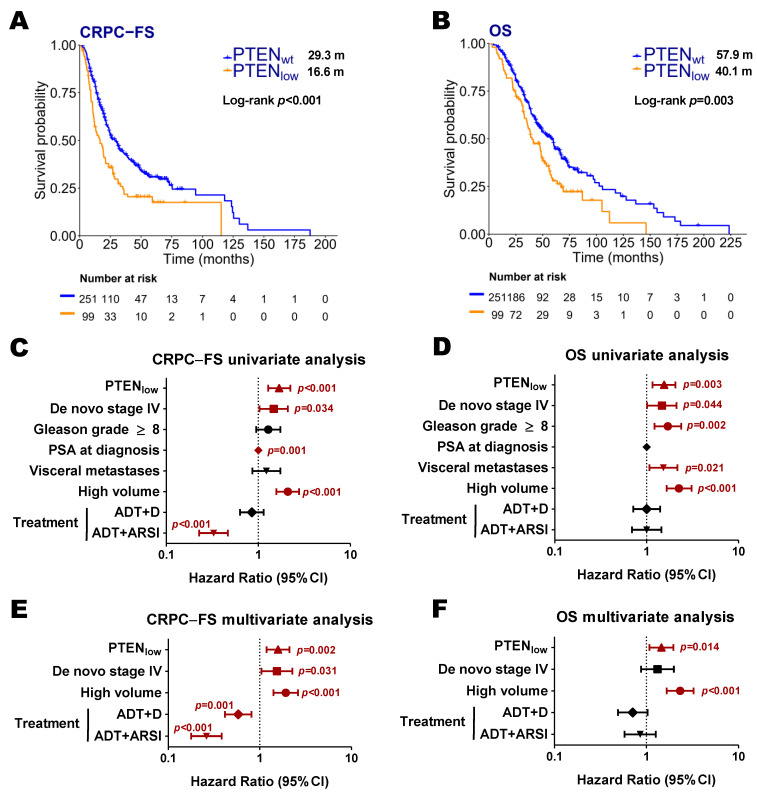
Clinical outcomes according to *PTEN* expression status. Kaplan–Meier curves representing CRPC-free survival (CRPC-FS) (**A**) and overall survival (OS) (**B**) according to *PTEN* expression (nCounter) in all patients; forest plots representing the univariate (**C**,**D**) and multivariate (**E**,**F**) analysis for CRPC-FS and OS in all patients. ADT: androgen deprivation therapy; ARSI: androgen receptor signaling inhibitors; CI: confidence interval; D: docetaxel; m: median months. Significant *p* values (*p* < 0.05) are indicated in bold.

**Figure 2 ijms-26-06244-f002:**
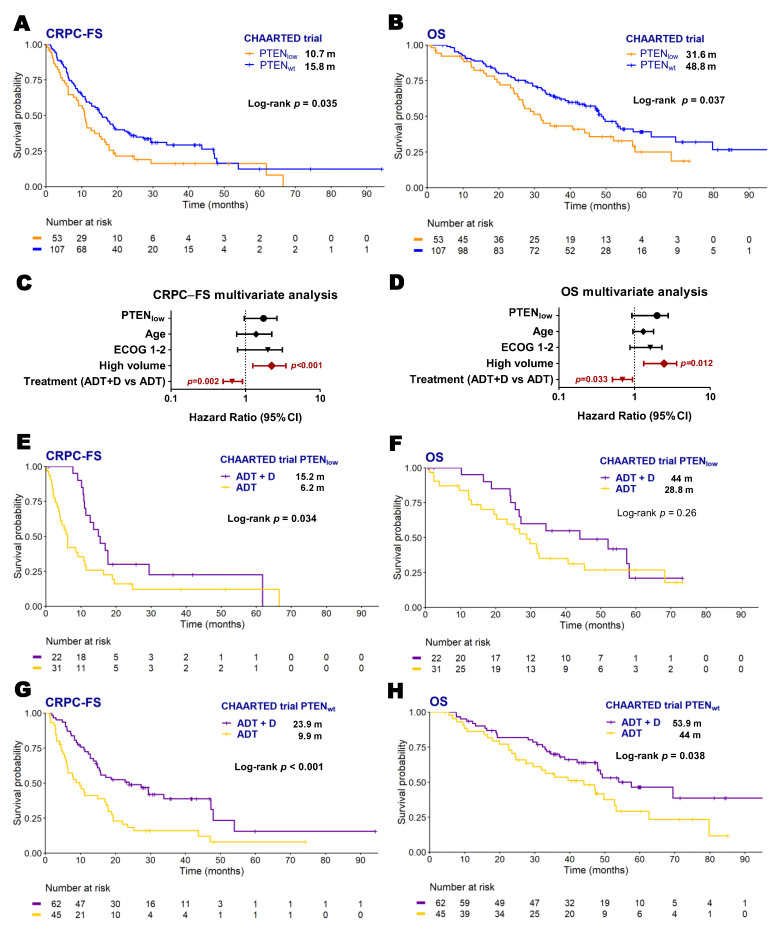
Clinical outcomes according to *PTEN* expression status in the CHAARTED trial patients. Kaplan–Meier curves representing CRPC-free survival (CRPC-FS) (**A**) and overall survival (OS) (**B**) according to *PTEN* expression of microarray data from the CHAARTED trial; forest plots representing the multivariate analysis for CRPC-FS (**C**) and OS (**D**) in the CHAARTED trial patients; Kaplan–Meier curves representing CRPC-FS (**E**,**G**) and OS (**F**,**H**) for patients from the CHAARTED trial according to *PTEN* expression segregated by treatment arm. ADT: androgen deprivation therapy; CI: confidence interval; D: docetaxel; ECOG: Eastern Cooperative Oncology Group; m: median months; significant *p* values (*p* < 0.05) are indicated in bold.

**Figure 3 ijms-26-06244-f003:**
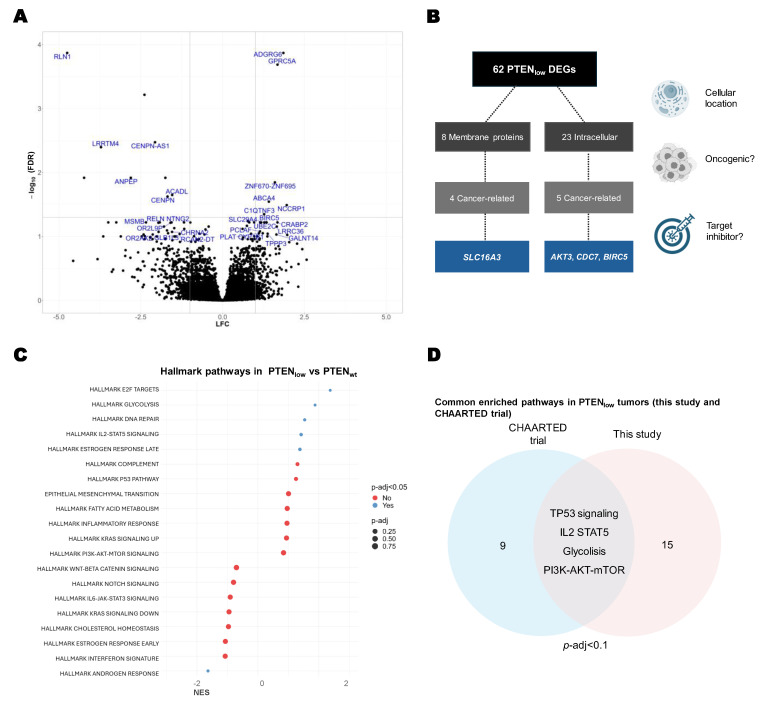
Transcriptional characterization of PTEN_low_ tumors. (**A**) Volcano plot of differential expressed genes (DEGs) in PTEN_low_ tumors vs. PTEN_wt_ (FDR < 0.1); gene names in blue indicate DEGs with a significant FDR < 0.05; (**B**) flowchart diagram of DEGs in PTEN_low_ tumors, according to cellular localization and available inhibitors; (**C**) gene set enrichment analysis (GSEA) of hallmark MSigDB pathways in the PTEN_low_ tumors vs. PTEN_wt_ (RNA-Seq, *n* = 60); (**D**) Venn diagram of the overexpressed pathways found in this study and the CHAARTED trial microarrays (*p*-adj < 0.1) from GSEA of hallmark MSigDB pathways in the PTEN_low_ tumors vs. PTEN_wt_.

**Figure 4 ijms-26-06244-f004:**
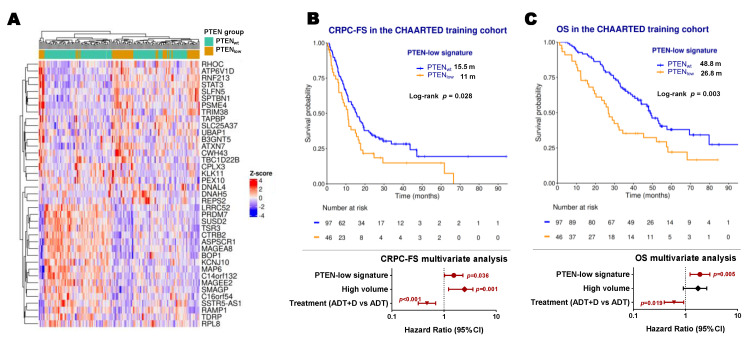
PTEN-low related signature. (**A**) Expression heatmap of the PTEN-low related signature genes in microarray data from the CHAARTED trial; Kaplan–Meier representing CRPC-free survival (CRPC-FS) (**B**) and overall survival (OS) (**C**) and forest plots representing the multivariate analysis according to classification by the PTEN-low signature in the training cohort of CHAARTED trial patients. ADT: androgen deprivation therapy; CI: confidence interval; D: docetaxel; m: median months; significant *p* values (*p* < 0.05) are indicated in bold.

**Figure 5 ijms-26-06244-f005:**
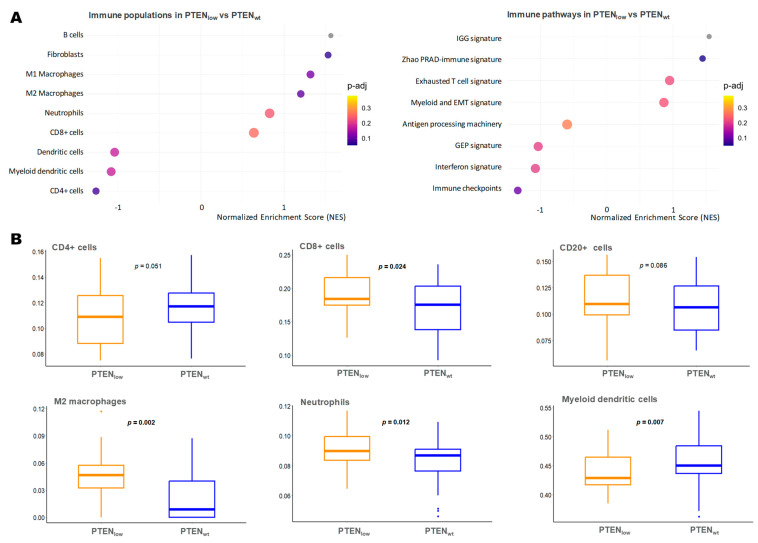
Immune microenvironment characterization of tumors according to *PTEN* expression status. (**A**) Gene set enrichment analysis (GSEA) of immune microenvironment populations and signatures in the PTEN_low_ tumors vs. PTEN_wt_ (RNA-Seq); (**B**) boxplots of the most relevant immune cell signatures expression (ssGSEA score) in PTEN_low_ tumors vs. PTEN_wt_ (Wilcoxon-test *p*-value); EMT: epithelial–mesenchymal transition; GEP: T-cell inflamed gene expression profile; IGG: 14-gene immunoglobuline signature; M2 Macrophages: macrophages with M2 differentiation over M1; PRAD: prostate adenocarcinoma.

**Figure 6 ijms-26-06244-f006:**
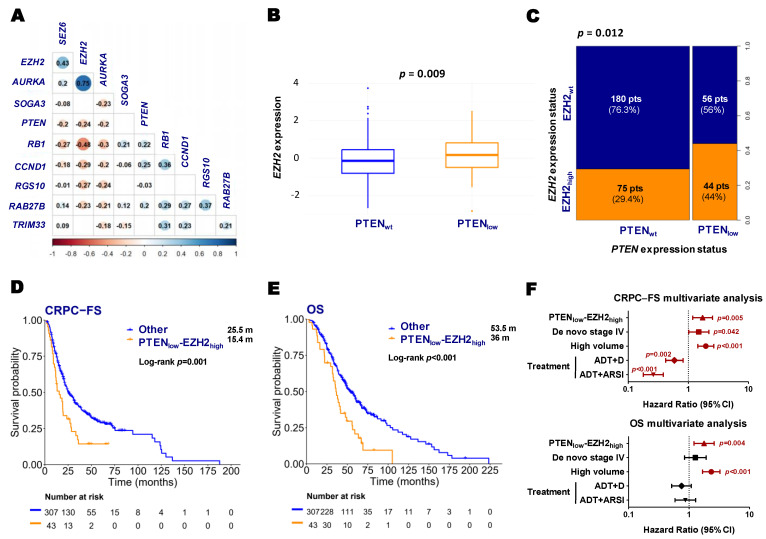
Neuroendocrine genes and clinical outcomes according to *PTEN* and *EZH2* expression status. (**A**) Correlation matrix between *PTEN* and neuroendocrine genes from RNA-Seq expression values (correlation coefficient (r) and colored dots if *p* < 0.05); (**B**) boxplot of *EZH2* expression according to *PTEN* status (nCounter) (Wilcoxon test *p*-value); (**C**) spine plot representing the proportion of patients according to *PTEN* and *EZH2* expression status (nCounter) (Fisher’s exact test *p*-value); Kaplan–Meier curves representing CRPC-free survival (CRPC-FS) (**D**) and overall survival (OS) (**E**) of all patients according to PTEN_low_-EZH2_high_ expression (nCounter); (**F**) forest plots representing the multivariate analysis for CRPC-FS and OS in all patients. ADT: androgen deprivation therapy; ARSI: androgen receptor signaling inhibitors; CI: confidence interval; m: median months; pts: patients. Significant *p* values (*p* < 0.05) are indicated in bold.

**Table 1 ijms-26-06244-t001:** Characteristics of all patients (ADT, ADT + D, and ADT + ARSI cohorts) and all patients segregated according to PTEN expression.

	All Patients	PTEN_low_	PTEN_wt_	*p*-Value
Patients, *n* (%)	355	100 (28.2)	255 (71.8) *	
Age (years)				
Median (range)	68.1 (46.3–92.8)	69.7 (47.7–86.5)	67.8 (46.3–92.8)	0.456
PSA at diagnosis (ng/mL)				
Median (range)	40.3 (0.02–7448)	32.5 (1.8–4860)	42 (0.02–7448)	0.452
ECOG performance status score, *n* (%)				
0	135 (38)	35 (35)	100 (39.2)	0.542
1 or 2	196 (55.2)	61 (61)	135 (52.9)	
NA	24 (6.8)	4 (4)	20 (7.8)	
Stage at diagnosis, *n* (%)				
<IV	70 (19.7)	16 (16)	54 (21.2)	0.3
IV	271 (76.3)	80 (80)	191 (74.9)	
NA	14 (3.9)	4 (4)	10 (3.9)	
Gleason sum at diagnosis, *n* (%)				
≤7	84 (23.7)	20 (20)	64 (25.1)	0.402
≥8	261 (73.5)	76 (76)	185 (72.5)	
NA	10 (2.8)	4 (4)	6 (2.4)	
Presence of visceral metastases, *n* (%)				
Yes	55 (15.5)	20 (20)	35 (13.7)	0.142
No	296 (83.4)	78 (78)	218 (85.5)	
NA	4 (1.1)	2 (2)	2 (0.8)	
Disease volume, *n* (%)				
High	221 (62.3)	64 (64)	157 (61.6)	0.537
Low	129 (36.6)	33 (33)	96 (37.6)	
NA	5 (1.4)	3 (3)	2 (0.8)	

ADT: androgen deprivation therapy; ARSI: androgen receptor signaling inhibitor; D: docetaxel; ECOG: Eastern Cooperative Oncology Group; *n*: number of cases; NA: not available; PSA: prostate-specific antigen. *p*-value is based on Fisher exact test and Wilcoxon Mann–Whitney test for categorical and continuous variables, respectively. * Five patients were excluded from survival analysis due to lack of complete follow-up data.

## Data Availability

The Nanostring nCounter gene expression data presented in this study are available in Appendix A. RNA-Seq data presented in this study are available on reasonable request from the corresponding author and subject to authorization by the institution. Microarray expression data from patients included in the CHAARTED clinical trial deposited in the GEO database with the accession number GSE201805 [19] were used for the independent validation analysis.

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
