# Peer review of "Clinical and Transcriptomic Characterization of Metastatic Hormone-Sensitive Prostate Cancer Patients with Low PTEN Expression"

_ijms, 2025, doi:10.3390/ijms26136244_

Round 1
Reviewer 1 Report
Comments and Suggestions for Authors
This manuscript by Herreros et al. evaluates the prognostic value of low-level PTEN mRNA expression in 350 men with metastatic hormone-sensitive prostate cancer. PTEN-low tumors, which made up 28% of cases, were independently associated with markedly shorter time to castration-resistant disease and overall survival across all first-line regimens examined in the study.
RNA-seq profiling of 60 tumors showed that PTEN-low cancers preferentially up-regulate PI3K/AKT/mTOR, glycolytic, DNA-repair and inflammatory programs while suppressing androgen-receptor signaling. An elastic-net-derived 39-gene PTEN-low signature predicted poorer outcomes in the CHAARTED microarray cohort but performed only modestly in an independent RNA-seq validation set. Immune-deconvolution suggested a mixed infiltrate of neutrophils, M2 macrophages and exhausted CD8+ T cells.
Co-over-expression of EZH2 in PTEN-low samples defined an ultra-poor-prognosis subgroup.
Major concerns
- The analysis presented by the authors is inherently limited by its retrospective, non-randomized design and substantial treatment heterogeneity; despite multivariable modelling, residual confounding across the three first-line therapy groups may inflate hazard-ratio estimates. The authors should acknowledge the limitation of their work.
- The authors on a single, previously defined PTEN mRNA cut-off raises questions about its analytical robustness across different platforms and populations. The authors should address this limitation.
- In Figure 4, the authors propose a 39-gene PTEN-low signature, external validation is weak. Gene panel's prognostic performance falls sharply in the independent RNA-seq cohort. So the clinical utility of such a gene panel remains unproven.
- The authors should make the data publicly available.
Minor concerns
- The authors should use adjusted p-values (after multiple test correction) in Figure 3A volcano-plot.
- Figures 3C and D have been labelled as A and B in the main figure.
Reviewer 2 Report
Comments and Suggestions for Authors
highly interesting manuscript on one of the most promising directions in PC biomarkers research - PTEN tumor suppressor gene.
title - clearly defining the topic of the manuscript - No remarks
Introduction - in-depth and systematic analysis of the contemporary literature
row 88-91 - this statement needs a reference - Minor
Results - properly visualized and firmly establishing the authors` conclusions
Figure 1 C - PSA= 0.001 ? - discrepancy between the figure and caption? - Minor
Discussion and conclusions - elegantly implementing their results into the contemporary literature, the authors firmly established their conclusions - No remarks
material and methods - the main benefit of the study is highly rigorous and sophisticated study protocol - PTEN expression in own and validation cohort, transcriptional characterization of PTENlow Tumors and development of PTENlow related signature, as well as additional research on neuroendocrine and immune microenviroment related genes - No remarks
